# Subjective data, objective data and the role of bias in predictive modelling: Lessons from a dispositional learning analytics application

**Dirk Tempelaar**[1]*, **Bart Rienties**[2]ᵒ, **Quan Nguyen**[3]ᵒ

**1** School of Business and Economics, Maastricht University, Maastricht, The Netherlands, **2** Institute of Educational Technology, Open University UK, Milton Keynes, United Kingdom, **3** School of Information, University of Michigan, Ann Arbor, MI, United States of America

ᵒ These authors contributed equally to this work.
* D.Tempelaar@MaastrichtUniversity.nl

**Data Availability Statement:** The data, the MPlus and SPSS codes, and the main components of the output are archived in DANS, the Data Archiving and Networked Services of the NOW, the Dutch

## Abstract

For decades, self-report measures based on questionnaires have been widely used in educational research to study implicit and complex constructs such as motivation, emotion, cognitive and metacognitive learning strategies. However, the existence of potential biases in such self-report instruments might cast doubts on the validity of the measured constructs. The emergence of trace data from digital learning environments has sparked a controversial debate on how we measure learning. On the one hand, trace data might be perceived as "objective" measures that are independent of any biases. On the other hand, there is mixed evidence of how trace data are compatible with existing learning constructs, which have traditionally been measured with self-reports. This study investigates the strengths and weaknesses of different types of data when designing predictive models of academic performance based on computer-generated trace data and survey data. We investigate two types of bias in self-report surveys: response styles (i.e., a tendency to use the rating scale in a certain systematic way that is unrelated to the content of the items) and overconfidence (i.e., the differences in predicted performance based on surveys' responses and a prior knowledge test). We found that the response style bias accounts for a modest to a substantial amount of variation in the outcomes of the several self-report instruments, as well as in the course performance data. It is only the trace data, notably that of process type, that stand out in being independent of these response style patterns. The effect of overconfidence bias is limited. Given that empirical models in education typically aim to explain the outcomes of learning processes or the relationships between antecedents of these learning outcomes, our analyses suggest that the bias present in surveys adds predictive power in the explanation of performance data and other questionnaire data.

## Introduction

SBBG or the 'Snapshot, Bookend, Between-Groups' paradigm is the unflattering description by Winne and Nesbit [1] of the current state of affairs of building and estimating educational models.

organization of scientific research. DANS is an open access resource. The final version of this archive, labelled Tempelaar, D, 2020, "Replication Data for PlosOne 2020 manuscript Tempelaar ea", has received the unique handle: https://hdl.handle.net/10411/YAF7CJ, DataverseNL

**Funding:** The author(s) received no specific funding for this work.

**Competing interests:** The authors have declared that no competing interests exist.

The data reference in that description is provided by the snapshot 'S', an S that could equally well stand for a survey or self-report. In the alternative paradigm of a 'more productive psychology of academic achievement' [1, p. 671] that the authors offer, the use of trace data collected over time that describe learning episodes, supplemented with some snapshot data represents one of the paradigmatic changes suggested. Other researchers go even further in restricting the role of snapshot type of data in educational research. For example, in line with traditions in the area of metacognition terming the different data paradigms as off-line and online, Veenman [2, 3] limits the description of the properties of off-line data to a list of 'fundamental validity problems', such as the problem of the individual reference point (variability in perspective chosen by learners), memory problems (failing to correctly retrieve past experiences), and the prompting effect problem (item steering in a direction different from what a spontaneous self-report will bring). If these observations were to be representative for the development of empirical educational research, it suggests that research papers based on questionnaire data do not have such a bright future.

Reading through this methodological litany, the asymmetry in descriptions of data sources and data types in educational research stands out. Where the fundamental validity problems going with questionnaire data are typically spelt out at considerable detail, with the above issues raised by Veenman [2, 3] being no more than the top of the iceberg, a critical evaluation of the characteristics of online data, or trace data, is often missing. It is as if online data represents by definition "true", unbiased data, which are both valid and reliable [2–4]. Much to our surprise, in the research area of learning analytics, an opposite development can be observed. Several learning analytics researchers explicitly recognize the limitations of models designed on online data only [5,6]. There is an emergence of research that seeks to integrate different types of data, such as visible from the new area of dispositional learning analytics that searches to append trace data, the classical subject of learning analytics research, with questionnaire measured disposition data, or educational research using multi-modal data [5,6].

The aim of this study is to showcase the benefits of critically assessing the characteristics of trace and questionnaire data. This showcase is developed in the context of a dispositional learning analytics application that combines a wide variety of data and data types: trace data from technology-enhanced learning systems, computer log data of static nature, questionnaire data, and course performance data. In the survey research literature, it is widely acknowledged that although questionnaires and psychometric instruments measuring constructs like anxiety, motivation, or self-regulation, have strong internal and external validity, many respondents have a typical response style [7, 8]. For example, some learners are more inclined to have an acquiescence style of response (i.e., the tendency to yeah saying), while others tend to extreme responses (i.e., using the extremes on the Likert response scale). Similarly, in terms of confidence biases, some learners might underestimate their abilities, skills, and knowledge, while others might overestimate their confidence [9]. One view is to consider these response styles and confidence biases as unwelcome, another is that these "biases" could potentially be used as interesting proxies of underlying features of a respondent. Therefore, as an instrument to characterise this rich set of data, we develop two alternative approaches: one building on the framework of response styles; and an alternative one based on the difference between subjective and objective notions of confidence in one's learning.

Both response styles and confidence differences serve as a potential source of biases in the data. One would expect this to refer to self-report questionnaire data only, but we will investigate other data types (e.g., trace data, performance data) on the presence of these anomalies. Based on that analysis, we intend to answer two generalised research questions:

- In a data-rich context consisting of data of questionnaire type, trace data of both process and product type and performance data, how can we decompose each data element into a component that represents the contribution of biases, such as response style bias or overconfidence, and a component independent of biases?

- If our modelling endeavour aims to design models that help predict course performance or explain the relationship between student's characteristics that act as antecedents of performance, what lessons can be learned from these decompositions into bias and non-bias components?

First, we will introduce the reader to the three buildings blocks of our study: dispositional learning analytics, response styles, subjective and objective confidence measures. Second, we will investigate the presence of any response styles and confidence difference components in subjective questionnaire data, objective trace data, and learning outcomes data types, and discuss their implications.

## Three buildings blocks: Dispositional learning analytics, response styles and confidence measures

### Dispositional learning analytics

Dispositional learning analytics proposes a learning analytics infrastructure [10–12] that combines learning data, generated in learning activities through the traces of technology-enhanced learning systems, with learner data, such as student dispositions, values, and attitudes measured through self-report questionnaires [5]. The unique feature of dispositional learning analytics is in the combination of learning data with learner data: digital footprints of learning activities, as in all learning analytics applications, together with self-response questionnaire learner data. In [5, 13], the source of learner data is found in the use of a dedicated questionnaire instrument specifically developed to identify learning power: a mix of dispositions, experiences, social relations, values and attitudes that influence the engagement with learning. In our own dispositional learning analytic research [14–18], we sought to operationalise dispositions with the help of instruments developed in the context of contemporary social-cognitive educational research, as to make the connection with educational theory as strong as possible. Another motivation to select these instruments is that they are closely related to educational interventions. These instruments include:

- The expectancy-value framework of learning behaviour [19], encompassing affective, behavioural, and cognitive facets;

- The motivation and engagement framework of learning cognitions and behaviours [20] that distinguishes learning cognitions and learning behaviours of adaptive and maladaptive types;

- Aspects of a student approach to learning (SAL) framework: cognitive processing strategies and metacognitive regulation strategies, from Vermunt's [21] learning styles instrument, encompassing cognitions and behaviours (see also [22]);

- The control-value theory of achievement emotions, both about learning emotions of activity and epistemic types, at the affective pole of the spectrum [23–25];

- Goal setting behaviour in the approach and avoidance dimensions [26];

- Academic motivations that distinguish intrinsically versus extrinsically motivated learning [27].

The type of dispositional learning analytics models we have developed within the above theoretical frameworks fit in the current trend in educational research to apply multi-modal data analysis by combining data from a range of different sources. In our research, we invariably find that predictive modelling focusing on learning outcomes or dropout finds formative assessment data as its dominant predictor. However, these formative assessment data are often less timely than one would wish, for example, for doing educational interventions early in the course. The best timely prediction models we were able to design are typically dominated by trace data of product type (e.g. tool mastery scores) combined with questionnaire data, with secondary roles for trace data of process type (e.g. number of attempts to solve math exercise 21, number of assignments completed in week 4), due to its unstable nature [15–18].

## Response styles

Response styles refer to typical patterns in responses to Likert response scales questionnaire items [7, 8, 28, 29]. Although intensively investigated in marketing, cultural, and health studies, response styles went largely unnoticed in empirical educational research. Response styles are induced by the tendency of respondents to respond in a similar way to items, independent of the content of the item, such as yeah saying, or seeking for extreme responses. In the literature, nine common types of response styles are distinguished:

- Acquiescence Response Style, ARS: the tendency to respond positively

- Dis-Acquiescence Response Style, DARS: the tendency to respond negatively

- Net-Acquiescence, NARS: ARS-DARS

- MidPoint Response Style, MRS: the tendency to respond neutrally

- Non-Contingent Response, NCR: the tendency to respond at random

- Extreme Response Scale, ERS: the tendency to respond extremely

- Extreme Response Scale, ERSpos and ERSneg: the tendency to respond extremely positively or extremely negatively

- Response range, RR: the difference between the maximum and minimum response

- Mild Response Style, MLRS: the tendency to provide a mild response.

Longitudinal research into the stability of response styles concludes that response styles function as relatively stable, individual characteristics that can be included as control variables in the analysis of questionnaire data [29]. Largest effects were found in studies of the ERS style [30], but explained variation never exceeded the level of 10%. Other empirical studies, such as [28, 31] focussed on the ERS only. Response styles constitute a highly collinear set of observations, by definition: for example, mild responses are the complement of extreme responses. Therefore, any analysis of response styles has to be based on a selection from the above styles.

In the fifties and sixties, response styles research focused on a second antecedent of response styles beyond personality: the domain of the questionnaire. The leading research question in those investigations was if response style findings can be generalised over different instruments. Findings indicate that this generalisation is partial: response styles contain both a generic component and an instrument-specific component [30, 32]. Empirical research is however limited in most cases to the comparison of response styles in two or three instruments; it is only in applications of dispositional learning analytics as in the current study that one can investigate commonalities in response styles over a broad range of instruments.

## Confidence measures

As a second source of response bias, we sought for an indicator of under- or overconfidence, or the difference between a subjective confidence measure and an objective one. Different operationalizations of this can be found, such as judgements of learning, feeling of learning, or ease of learning judgements [9]. Our operationalization of subjective confidence is best interpreted as a prospective, ease-of-learning indicator. It is based on an expectancy-value framework oriented survey [33] administered at the start of a course that generates several expectancy scores (such as perceived cognitive competence, or the expectation not to encounter difficulties in learning), and personal value scores. Subjective confidence is then defined as the predicted value of a learning outcome based on survey responses (e.g. the regression of exam performance for mathematics on the scores of the several expectancy- and value-constructs). A similar procedure can be applied to define objective confidence, whereby we define the predicted value of the regression of the exam score on two objective predictors available at the start of the course: the level of prior education and the score on a diagnostic entry test. The difference between these two regression-based predictions is seen as the difference of subjective and objective confidence or a measure of subjective overconfidence. The variables used in the calculations of response styles and the confidence difference will be described in the next section.

## Research methods

Ethics approval was obtained by the Ethical Review Committee Inner City faculties of Maastricht University (ERCIC_044_14_07). Participants of the research all provided written consent.

## Context of the empirical study

This study took place in a large-scale introductory mathematics and statistics course for first-year undergraduate students in a business and economics program in the Netherlands. The educational system is best described as 'blended' or 'hybrid' [34]. The main component is face-to-face: Problem-Based Learning (PBL), in small groups (14 students), coached by a content expert tutor (see [35] for further information on PBL and the course design). Participation in tutorial groups is required. Optional is the online component of the blend: the use of the two e-tutorials—SOWISO and MyStatLab (MSL) [18]. This design is based on the philosophy of student-centred education, placing the responsibility for making educational choices primarily on the student. Since most of the learning takes place during self-study outside class through the e-tutorials or other learning materials, class time is used to discuss solving advanced problems. Thus, the instructional format is best characterized as a flipped-classroom design [35].

The student-centred nature of the instructional design requires, first and foremost, adequate actionable feedback to students so that they can appropriately monitor their study progress and topic mastery. The provision of relevant feedback starts on the first day of the course when students take two diagnostic entry tests for mathematics and statistics, the mathematics test based on a validated, nation-wide instrument. Feedback from these entry tests provides a first signal for the importance of using the e-tutorials. Next, the e-tutorials take over the monitoring function: at any time, students can see their performance in the practice sessions, their progress in preparing for the next quiz, and detailed feedback on their completed quizzes, all in the absolute and relative (to their peers) sense. Students receive feedback about their learning dispositions through a dataset containing their personal scores on several instruments, and aggregate scores. These datasets are the basis of the individual student projects students do in the second last week of the course, in which they statistically analyse and interpret their

personal data and compare it with class means. Profiting from the intensive contact between students and their tutors of the PBL tutorial groups, learning feedback is directed at students and their tutors, who carry first responsibility for pedagogical interventions.

The subject of this study is the full 2018/2019 cohort of students, i.e. all students who enrolled in the course and administered the learning dispositions instruments: in total, 1080 students (that includes all first-year students, since the student project is a required assignment, but excludes repeat students, who did the project the previous year). A large diversity in the student population was present: only 21.6% were educated in the Dutch high school system. The largest group, 32.6% of the students, followed secondary education in Germany, followed by 20.8% of students with Belgian education. In total, 57 nationalities were present. A large share of students was of European nationality, with only 4.8% of students from outside Europe. High school systems in Europe differ strongly, most particularly in the teaching of mathematics and statistics. For example, the Dutch high school system has a strong focus on the topic of statistics, whereas statistics are completely missing in high school programs of many other European countries. Next, all countries distinguish different tracks of mathematics education at the secondary level, with 31.5% of our students educated at the highest, advanced level preparing sciences, and 68.5% of students educated at the intermediate level, preparing social sciences. Therefore, it is crucial that this present introductory module is flexible and allows for individual learning paths, which is the reason to opt for a blended design with providing students with a lot of learning feedback generated by the application of dispositional learning analytics [18, 35].

## Instruments and procedure

In this study, we combine data of different types: course performance measures, Learning Management System (LMS) and e-tutorial trace variables, Students Information System (SIS) based variables, and learning disposition variables measured by self-report questionnaires. As suggested by Winne's taxonomy of data sources [4, 36, 37], our study applies self-report questionnaire data and trace data through the logging of study behaviours and the specific choices students make in the e-tutorials.

The self-report questionnaires applied in this study are described in S1 Appendix: achievement emotions (A. 1), epistemic emotions (A. 2), achievement goals (A. 3), motivation and engagement (A. 4), attitudes towards learning (A. 5), approaches to learning (A. 6) and academic motivations (A. 7). These questionnaires are all long-existing instruments, well-described, and validated in decades of empirical research into educational psychology. Most were administered in the first two weeks of the course, at different days, each administration taking between five and ten minutes. The first exception is the instrument quantifying emotions by participating in learning activities (described in section A. 1), which was administered halfway through the course. This was done to allow students sufficient experiences with the learning activities, while simultaneously avoiding the danger that an approaching exam might strongly impact learning emotions. A second exception is that the motivation and engagement instrument (described in section A. 4), was administered twice: at the start and the end of the course (T2). Since data from the self-report questionnaires are used by the students in individual statistical projects that analyse personal learning data, the responses cover all students (except for about 15 students dropping out). To ease the administration of the questionnaires, all applied the same response format of a seven-point Likert scale. Students provide consent that their personal data is used outside the project for learning analytics-based feedback and educational research.

**Course performance measures.** The final course performance measure, Grade, is a weighted average of final exam score (87%) and quiz scores (13%). Performance in the exam

has two components with equal weight: exam score mathematics (MathExam) and exam score statistics (StatsExam). The same decomposition refers to the aggregated performance in the quizzes for both topics: MathQuiz and StatsQuiz.

**Trace data from technology enhancing learning systems.** Three digital systems have been used to organise the learning of students and to facilitate the creation of individual learning paths: the LMS BlackBoard and the two e-tutorials SOWISO for mathematics and MSL for statistics. From the BlackBoard trace variables, all of the process type, based upon our previous research, we choose BBClicks as the total number of clicks in BlackBoard. From the thousands of trace variables available from the two e-tutorial systems, we selected one product type variable and a few process type variables, all on an aggregate level. The product variable represents mastery achieved in the e-tutorials, as the proportion of exercises correctly solved: MathMastery and StatsMastery. Main process type of variables are the number of attempts to solve an exercise, totalled over all exercises: MathAttempts and StatsAttempts, and total time on task: MathTime and StatsTime. Next, the Sowiso system archives the feedback strategies students apply in solving any exercise, resulting in additional process variables MathHints, the total number of hints asked for, and MathSolutions, the number of worked-out examples asked for.

**SIS system data and entry tests.** Our university SIS provided several further variables mainly used for control purposes. Standard demographic variables are Gender (with an indicator variable for female students), International (with an indicator for non-Dutch high school education), and MathMajor (with an indicator for the advanced mathematics track in high school). The MathMajor indicator is constructed based on distinguishing prior education preparing for either sciences or social sciences. Finally, students were required upon entering the course to complete two diagnostic entry tests, one for mathematics (MathEntry), and one for statistics (StatsEntry).

## Data analysis

The data analysis of this study contained a sequence of steps. In several of these steps, different options were available as to how to proceed in the analysis. We will shortly explain the choices we made, without suggesting that other choices cannot work as well. In fact, this study would lend itself to an application of 'multiverse analysis' [38]: performing the analyses across a set of alternative data sets applying alternative statistical methods to find out how robust the empirical outcomes are.

Our dispositional learning analytics-based dataset consisted of several types of data: self-report questionnaire data as the dispositions, trace data from learning enhancing systems, demographic data from SIS type of systems, and course performance data.

All questionnaires were administered with items of the Likert 1...7 type, to simplify the response by students. Since the different instruments applied different labels for the several Likert options, we used the three anchors as labels: the negative pole, the neutral anchor and the positive pole. The 7-point Likert scale is a relatively long scale where most response style literature is based on 4-point or 5-point Likert scales. The use of this 7-point Likert scale, as well as the large size of our sample, comply with the outcomes of a recent simulation study [39] that signals a loss of control of Type 1 error for scales shorter than 7-point and samples smaller than 100.

Researchers investigating very long scales (9-, 10- or 11-point scales) have applied alternative operationalisations of extreme responses, including two extreme response categories at each end of the continuum [32]. Being in between those short and long scales, we opted to analyse both cases defining extreme responses as the proportion of responses in the single most extreme category as well as the proportion of responses in the two most extreme categories.

That is: we defined extreme negative response as the proportion of responses equal to 1 (ERSneg1) or equal to 1 or 2 (ERSneg2), and we defined extreme positive response as the proportion of responses equal to 6 or 7 (ERSpos2) or equal to 7 only (ERSpos1). Analyses were performed for both operationalisations, but reporting was restricted to the case of defining extremity by two categories. An important reason to do so was based on distributional properties of the data: where measures of extreme responses based on the single most extreme outcome are strongly right-skewed, measures based on 1, 2 or 6, 7 together are only moderately skewed. Thus, to prevent the need of data transformations that would make an interpretation of the outcomes of the regression models less straightforward, we opted for the current operationalisation. Next: preliminary analysis suggested that the effects of extreme responses depend on the direction: positive or negative. So, whereas most empirical studies in response styles aggregate positive and negative extreme responses into one category [29, 31], we chose to differentiate the two directions. As the Results section will indicate, in most of the models, we found that positive and negative extreme responses had opposite effects, suggesting that aggregation into the total extreme response is dubious.

We used response styles as one approach to operationalising bias. A set of 13 response styles was calculated for all eight questionnaire administrations: ARS, ARSW, DARS, DARSW, MRS, NARS, NARSW, RR, NCR, ERSneg1, ERSpos1, ERSneg2, and ERSpos2, where the last four styles were described above: negative and positive extreme responses, and taking one or two response categories into account. By definition, this set of response styles was strongly collinear, making a selection necessary. We followed other empirical studies in this area [31] by focusing on only ERS as a descriptor of response styles since the style was found to be relatively stable in repeated measurements and in this way acted as a personality characteristic [29]. This constitutes the response style that had the strongest impact on measures of central tendency for questionnaire scales, thus the strongest bias.

After computing response style measures for each of the eight questionnaire administrations, we investigated stability over different questionnaire instruments and calculated aggregated measures of response styles. In that aggregation, we excluded the second, end of course administration of the motivation and engagement instrument, so that aggregated measures represented averages of response styles from seven different instruments. An advantage of keeping one instrument apart was that it allowed investigating the role of both stability and endogeneity (the external validation of our extreme response measures, by investigating their role in the explanation of responses to an instrument not included in the calculation of extreme response measures). Concerning endogeneity: if we analysed the role of an aggregate measure of response style and the outcomes of one survey, did it matter much if in the calculation of the aggregated measures we included or excluded the specific survey?

The seven instruments used to generate the aggregated response styles counted in total 77 scales. Of these scales, a majority of scales, 46, were of adaptive or positive valence (examples are the enjoyment of learning, study management, valuing university, intrinsic but also extrinsic motivation). A minority of scales, 13, were of maladaptive (hampering learning activity) or negative (unpleasant) valence (such as a-motivation, boredom, disengagement). The balance between positive or adaptive and negative or maladaptive items differed from instrument to instrument, thereby impacting response style measures, as described in the literature [40].

After estimating aggregated ERSpos and ERSneg levels for all students in the sample, and estimating the confidence difference level for all students in the sample, we applied an instrumental variables approach in the remainder of the study. All variables were regressed on the two sets of bias factors, allowing to decompose all variables into two, orthogonal components: the part of the variable explained by the bias (indicated with 'RS' for the response style bias, indicated with 'Conf' for the confidence difference bias) and the part that is left unexplained

(the residual of the regression, indicated with 'RScor' and 'Confcor' to address the bias-corrected measures). That decomposition is described by the beta weights. To provide an example: the AEQ variable learning anxiety(LAX), when regressed on the two selected response styles ERSpos and ERSneg, generated the following regression equation:

$$LAX = 0.238^{***} * ERSpos - 0.640^{***} * ERSneg, R^2 = 0.430$$

These regression outcomes were used in two different manners. First, the regression betas were taken as the variable specific response styles. Using the above example, we defined the positive extreme response of variable LAX as the beta weight of the variable ERSpos in the regression of LAX on ERSpos and ERSneg. In this example: ERSpos(LAX) = 0.238. Likewise, we defined the negative extreme response as the beta weight of ERSneg in the same regression; thus, ERSneg(LAX) = − 0.640. Note: since ERSpos and ERSneg are nearly orthogonal, these beta weights are approximately equal to the correlations of LAX with the two response styles. Table B1 in S2 Appendix provides an overview of all variables in this study, including values of ERSpos, ERSneg and ΔConfidence calculated as beta weights.

The second way the above regression equation was used is to decompose LAX into the explained part of the regression equation, denoted as LAXRS since that explained part is a linear combination of the two response styles, and the residual part of the regression equation, denoted LAXRScor: the LAX score corrected for the response style. The beta weights of that decomposition are:

$$LAX = 0.656 * LAXRS + 0.755 * LAXRScor$$

The squares of these beta weights provide the contribution to explained variation: LAXRS explains $65.6^2$% = 43.0% of variation in LAX (the $R^2$ of the regression equation), LAXRScor explains $75.5^2$% = 57.0% of variation in LAX (due to orthogonality, the two percentages of explained variation sum to 100%).

In exactly the same manner, we constructed the ΔConfidence(LAX) score as the beta weight of the regression of LAX on ΔConfidence (see Table B1 in S2 Appendix; since this is a univariate regression, that beta weight equals the correlation) and we decomposed the variable LAX into a predicted and residual part using the variable ΔConfidence as an instrument. That decomposition is indicated as LAXConf and LAXConfcor. In these decompositions, LAXRS and LAXConf represent the bias components, and LAXRScor and LAXConfcor the de-biased, bias-corrected, components.

This procedure was applied to all variables under study, including the 'objectively' measured variables. That is, self-report constructs, trace variables of the process and product types, and course performance variables were all assigned variable specific scores for ERSpos, ERSneg and ΔConfidence, and were all decomposed into predicted and residuals components, both with response styles and ΔConfidence as instruments. For the self-report constructs, an alternative operationalisation of extreme responses would have been the extreme response scores of the items belonging to the specific scale. However, this procedure would have limited the analysis to the scale-based self-report variables only and would not allow for constructing an overconfidence component in the data; therefore, we opted for the above approach.

The last step in the analysis was to estimate models of educational processes, in three different modes:

• using only observed, uncorrected versions of the variables, resulting in traditional models based on observed data;

- using corrected, de-biased versions of the variables only, deriving alternative models that excluded biases resulting from response styles or confidence differences;

- the combined model, with observed, uncorrected response variables, and as explanatory variables the combination of corrected, de-biased versions of the survey or trace variables, together with the response style variables or the confidence difference variable.

In this third model, the bias terms are orthogonal to the bias-corrected variables, allowing quantifying the differential impact of response styles or confidence difference variables on models of educational processes. Models we estimated are all of the multiple regression types, for which IBM SPSS vs 26 was applied. Omega reliability measures were calculated in MPlus vs 8.4, using code developed by Bandalos [41, p. 396].

## Results

The several subsections will follow the sequential steps in the statistical analysis, described above. At first, we investigate response styles and confidence differences as sources of bias in questionnaire measurements in the first two subsections. All the following subsections document comparisons of models estimated with observed scores and models based on corrected scores using the instrumental variables approach. The first three subsections give insight into the outcomes of the decompositions of all variables under study. In the following subsection, we investigate the impact of these decompositions on the design and estimation of educational models. Given that we collected data based on a wide range of theoretical frameworks, a large number of different models can be estimated (and was indeed estimated). Our reporting is based on a, somewhat arbitrary, selection from all these models. In section four, we estimate the CVTAE model for achievement emotions. Section five investigates epistemic emotions as antecedents of achievement emotions. In the last two sections, we look into models that include other types of data than survey data only. In section six, we predict course performance variables from achievement emotions. And in section seven, we predict course performance variables from trace data. In all of these modelling endeavours, the main emphasis is on the role of the decomposition of all variables in bias and bias-corrected components.

## Response styles

Response styles of different instruments demonstrate some variation in descriptive values, as visible in Table 1, which can be explained by the balance between adaptive or positive items in the instrument at the one side, and negative or maladaptive items at the other side (in line with findings of other research, [33]). The AEQ instrument has lowest ARS and ERSpos scores. At the same time, the AEQ has the highest proportion of negatively valenced items (44 out of 54 items, or 81%) and the highest proportion of maladaptive items (33 out of 54 items; or 61%; the eleven learning anxiety items are negatively valenced but of adaptive type). Likewise, EES has a majority of negatively valenced items. Students tend to disagree with these negatively valenced or maladaptive items, causing lower ARS and ERSpos scores, and higher DARS and ERSneg scores. In contrast, the AGQ contains only positively valenced items, and only adaptive or neutral items (depending on how one classifies items with a performance valence). AGQ has the highest ARS and ERSpos scores, the lowest DARS and ERSneg scores.

Scale reliabilities have been estimated by two different measures: Cronbach's alpha measure and the omega measure. Omega measures have the advantage over alpha measures that they do not require the strict assumptions that come with the alpha measures and are violated in many situations [42]. Omega measures are calculated in MPlus based on code described in [41].

**Table 1. Descriptive statistics of response styles measures and the overall mean, median, reliability measures alpha and omega and skewness of all response styles.**

|  | Mean | Me-dian | MES | EES | AGQ | ILS | AEQ | AMS | SATS | MESt2 | Alpha | Omega | Skew-ness |
|---|---|---|---|---|---|---|---|---|---|---|---|---|---|
| ARS | 0.56 | 0.56 | 0.57 | 0.45 | 0.80 | 0.61 | 0.33 | 0.61 | 0.54 | 0.57 | 0.66 | 0.67 | -0.08 |
| DARS | 0.28 | 0.28 | 0.33 | 0.30 | 0.08 | 0.22 | 0.51 | 0.25 | 0.31 | 0.33 | 0.65 | 0.68 | 0.11 |
| MRS | 0.16 | 0.15 | 0.10 | 0.24 | 0.12 | 0.18 | 0.16 | 0.14 | 0.15 | 0.10 | 0.73 | 0.74 | 0.66 |
| MRLS | 0.49 | 0.49 | 0.34 | 0.66 | 0.32 | 0.63 | 0.54 | 0.52 | 0.45 | 0.41 | 0.80 | 0.80 | 0.06 |
| NARS | 0.28 | 0.28 | 0.24 | 0.15 | 0.73 | 0.39 | -0.18 | 0.36 | 0.23 | 0.24 | 0.64 | 0.65 | -0.08 |
| RR | 5.17 | 5.14 | 5.86 | 4.34 | 3.49 | 4.84 | 5.03 | 5.68 | 6.95 | 5.59 | 0.57 | 0.60 | -0.44 |
| NCR | 1.16 | 1.15 | 0.76 | 1.69 | 1.13 | 0.98 | 1.28 | 1.07 | 1.22 | 0.75 | 0.42 | 0.43 | 0.26 |
| ERSneg1 | 0.06 | 0.05 | 0.13 | 0.04 | 0.02 | 0.02 | 0.13 | 0.02 | 0.08 | 0.11 | 0.70 | 0.78 | 1.71 |
| ERSneg2 | 0.15 | 0.13 | 0.26 | 0.16 | 0.04 | 0.09 | 0.34 | 0.06 | 0.20 | 0.24 | 0.69 | 0.76 | 0.58 |
| ERSpos1 | 0.16 | 0.15 | 0.18 | 0.05 | 0.37 | 0.07 | 0.04 | 0.17 | 0.16 | 0.14 | 0.75 | 0.78 | 1.00 |
| ERSpos2 | 0.34 | 0.34 | 0.40 | 0.19 | 0.63 | 0.28 | 0.13 | 0.42 | 0.35 | 0.35 | 0.73 | 0.74 | 0.13 |

Mean, Median, Alpha and Omega are calculated based on seven instruments MES, EES, AGQ, ILS, AEQ and SATS, thus excluding MESt2. Skewness is calculated for the Mean value of all response styles.

At the same time, there is a reasonable amount of stability in the response style measures, except for the RR and NCR variables, over the instruments: Cronbach's alpha values vary from .64 to .80. Two of the response styles are strongly right-skewed: ERSneg1 and ERSpos1.

There exists collinearity amongst the set of response styles, resulting from the overlap in their definitions. E.g., ARS correlates .74 with ERSpos2, DARS correlates .79 with ERSneg2 (see S3 Appendix). Therefore, we make a selection from the full set of response styles, based on choices made in other research, reliability, and skewness scores. That selection is MRLS as a mild response, ERSneg2 as a negative extreme response, and ERSpos2 as a positive extreme response. Since MRLS is the complement of ERSpos2 and ERSneg2, we will report the latter two variables in the following sections (shortly addressed as ERSpos and ERSneg). ERSpos and ERSneg are, be it quite weakly, positively related, with r = .103 (*p* = .001).

## Confidence scores

Objective confidence scores are calculated as the predicted value of the regression equation explaining the mathematics exam score from the indicator variable MathMajor (indicating an advanced level of mathematics classes in high school) and MathEntry as the score in the mathematics entry exam, taken at the start of the course. The regression equation, in beta weights, reads:

$$ConfObj = 0.280^{***} * MathMajor + 0.258^{***} * MathEntry, R^2 = 0.187$$

Subjective confidence scores are calculated as the predicted value of the regression equation explaining the mathematics exam score from the expectancy and value variables of the SATS instrument, in line with the expectancy-value theory. That regression equation, in beta weights, reads:

$$ConfSubj = 0.265^{**} * CognComp - 0.068^{**} * NoDifficulty + 0.202^{***} * Affect + 0.040 * Value - 0.098^{**} * Interest, R^2 = 0.164$$

In this regression, cognitive competence as expectancy variable and the affective component of intrinsic valuing have the main impact on subjective confidence. Extrinsic valuing has a non-significant impact.

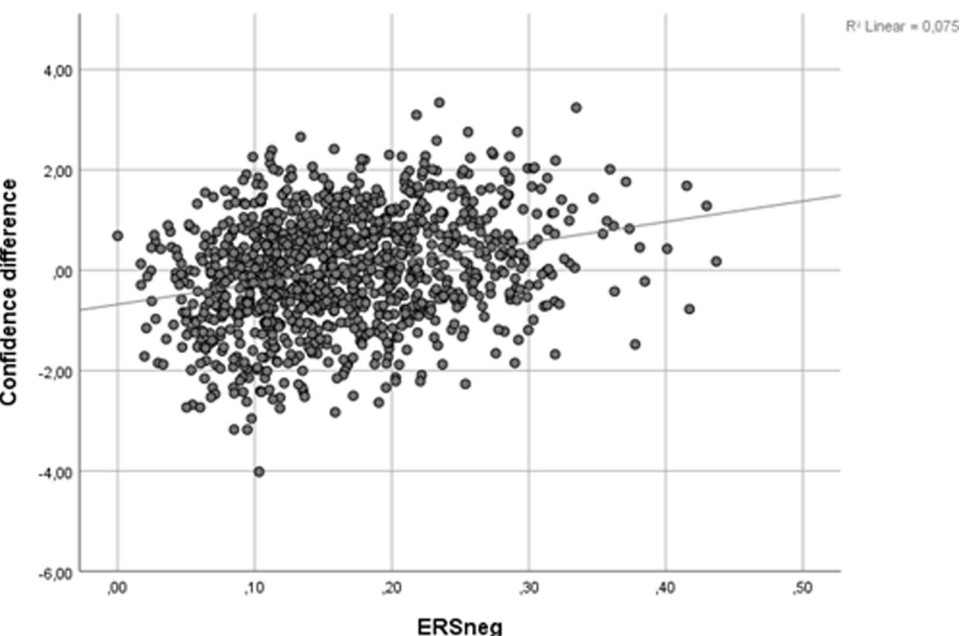

**Fig 1. Scatterplot of ΔConfidence against ERSneg, dots are representing students.**

ΔConfidence is defined as the difference between subjective and objective confidence. If objective confidence is regarded as the true level of confidence, it represents a measure of over-confidence. ΔConfidence is very weakly related to ERSpos ($r = -.066$, $p = .03$), but moderately positive related to ERSneg ($r = .273$, $p < .001$), as is visible from scatterplot presented in Fig 1, where each dot represents a student.

## Classification of variables based on response styles or overconfidence

The availability of response style measures allows new ways to categorise our data in educational studies. Rather than using the dichotomy of self-reported data versus objectively scored, we can position each variable of each data type in a two-dimensional plane of response styles: ERSpos and ERSneg. Fig 2 represents such a classification as a scatterplot, where each dot represents a variable in the analysis of either questionnaire, trace or learning outcome type. Variable numbers (see Table B1, S2 Appendix) are included in the scatter.

The distance of any point to the origin indicates how strong the role of response styles is in that variable. LAX (2), achievement anxiety, has the largest share of response styles in explained variation: 43%. LAX is characterised by large negative ERSneg score and modest positive ERSpos score. Other variables with that same characterisation are LHL (4), achievement helplessness, Anxiety (9), Confusion (8) and Frustration (10), the three epistemic emotions, and the three maladaptive motivations UC (29), FA (28) and AN (27): uncertain control, failure avoidance, and anxiety. In other words: these are negatively valenced, but mostly activating emotions, motivations and engagement variables.

The second group of variables positioned on the middle top of Fig 2 are characterised by high positive values of ERSneg, and small values of ERSpos. These variables are ASC (5), academic control, AB (32), academic buoyancy, and the several course performance variables: Grade (56), exam and quiz scores on both topics (57–60).

The largest cluster of variables has positive ERSpos scores and about zero ERSneg scores, as positioned on the right of Fig 2. These represent the several goal-setting behaviours (13–20),

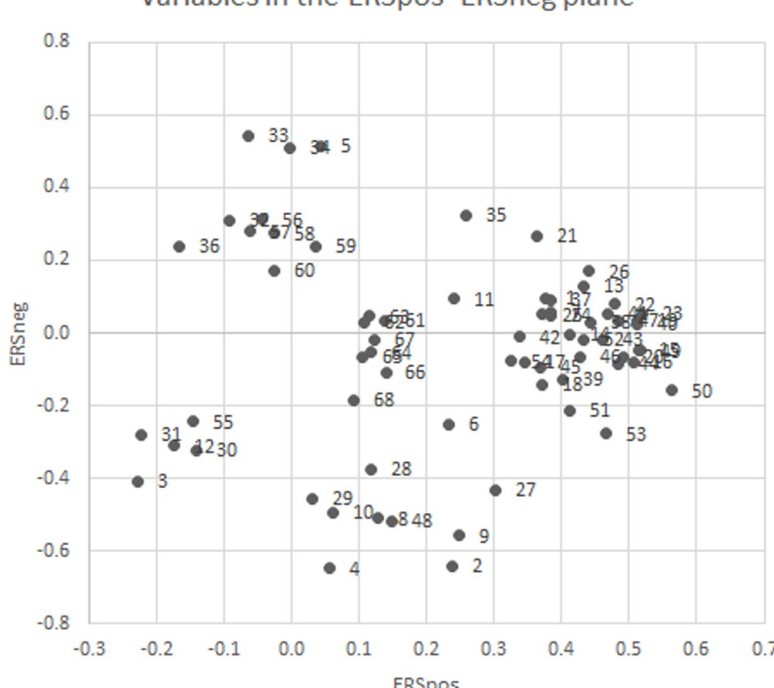

**Fig 2. Regression betas of ERSpos and ERSneg from regressions of educational variables on ERSpos and ERSneg.**

the cognitive and metacognitive scales (39–48), and academic motivation scales (49–54), all positively valenced scales. A smaller cluster is that of the trace variables (61–68), again with zero ERSneg scores, and small positive ERSpos scores positioned just to the right of the origin of the graph. The correlation between ERSpos and ERSneg, this time with variables as the subject, is nearly zero (r = .02).

A similar classification can be done for the confidence difference variable. Fig 3 provides the scatter of confidence difference against ERSneg. Low confidence difference observations are therefore epistemic emotions Anxiety (9), Confusion (8), Frustration (10), the three mal-adaptive motivations UC (29), FA (28) and AN (27): uncertain control, failure avoidance, and anxiety, and the two achievement emotions LAX (2) and LHL (4), learning anxiety and hope-lessness. These same variables make up the negative pole of ERSneg. At the other pole of the high positive confidence difference values, we find the variable AB (32), academic buoyancy, ASC (5), academic control, and SB (21), self-belief, that also distinguished in high positive ERSneg values. The correlation between ERSneg and confidence difference with the variables as the subject is high, .86 (much higher than the same correlation with students as subject). For that reason, Fig 3 is designed as the scatter of ΔConfidence against ERSneg (ΔConfidence is again no more than weakly related to ERSpos, with a correlation of -.18).

## The control-value theory of achievement emotions model

The CVTAE model is the simplest model to illustrate the suggested analytic approach within the current dataset. The model contains one predictor variable, academic control (ASC), and four response variables: learning anxiety (LAX), learning boredom (LBO), learning hopeless-ness (LHL) and learning enjoyment (LJO). The first step is to decompose all these five con-structs into a response style component and its residual, or a confidence difference component and it's residual. Table 2 provides that decomposition.

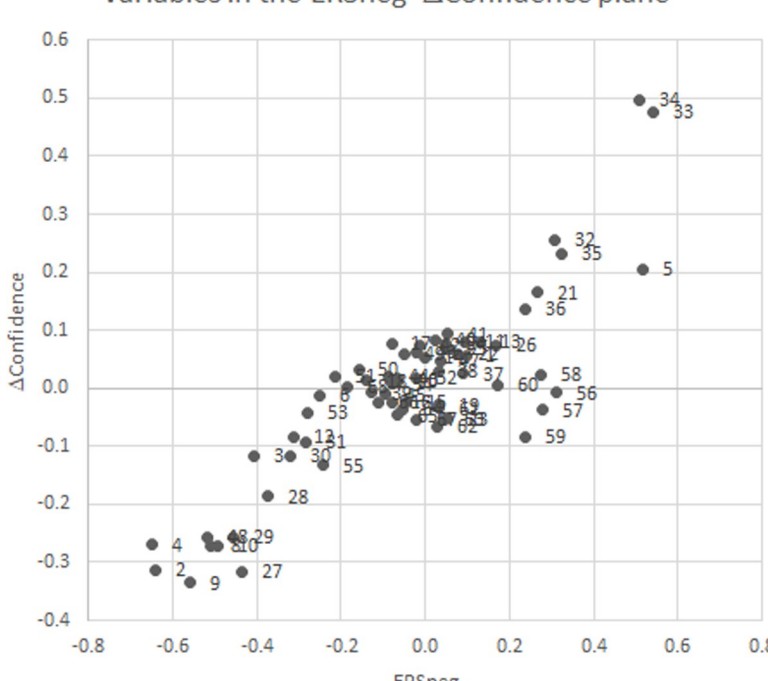

**Fig 3. Regression betas of ΔConfidence and ERSneg from regressions of educational variables on ΔConfidence respectively ERSpos and ERSneg.**

From the left panel of Table 2, we see that response styles account for a substantial amount of variation in the achievement emotions, up to more than 40% for anxiety and hopelessness. That is not the case in the right panel: explained variation by the confidence difference level is at a lower level, at most 10%. Left and right panel coincide concerning the ranking of the variables on explained variation: anxiety and hopelessness demonstrate the largest biases components, enjoyment the lowest. Given that ΔConfidence shares variation with ERSneg and ERSneg is the dominant predictor of anxiety and helplessness, this pattern is not surprising.

The four regression equations representing the CVTAE model estimated on observed values are contained in Table 3.

The same CVTAE model, now based on bias-corrected values, is provided in Table 4. Bias correction is based on response styles, left panel, or conference difference, right panel. Bias correction is applied to both left and right-hand side of the four regression equations, that is,

**Table 2. Decomposition of the AEQ variables.**

|  | ERSpos | ERSneg | $R^2$ | RS | RScor | ΔConf | $R^2$ | Conf | Confcor |
|---|---|---|---|---|---|---|---|---|---|
| ASC | .043 | .515*** | .272 | .522 | .853 | .204*** | .042 | .204 | .979 |
| LAX | .238*** | -.640*** | .430 | .656 | .755 | -.313*** | .098 | .313 | .950 |
| LBO | -.228*** | -.407*** | .240 | .490 | .872 | -.116*** | .013 | .116 | .993 |
| LHL | .058* | -.648*** | .415 | .644 | .765 | -.270*** | .073 | .270 | .963 |
| LJO | .376*** | .096*** | .159 | .399 | .917 | .055 | .003 | .055 | .998 |

columns two, three and four provide the regression outcomes of the variables in column one on the two response styles: standardised regression coefficients and explained variation. Columns five and six provide the beta weights of the response style-based decomposition. In the second panel, the regression on ΔConfidence, and the decomposition based on the confidence difference is provided.

e.g., anxiety corrected for response styles is regressed on academic control corrected for response style in the upper left panel.

The effects visible in the two panels are quite different. In the right panel, we see that correcting for overconfidence has an only limited impact: regression betas and explained variation decrease somewhat, but not much. The left panel shows a different picture. Most of the explanatory power is taken out by response style correction of anxiety and boredom values, together with ASC. In the case of boredom, academic control has even lost all of its explanatory power.

The last step in the analysis combines the corrected version of ASC with the bias terms, either ERSpos and ERSneg or ΔConfidence, as predictors of the four observed learning emotion variables. In Table 5 the outcomes of this last step are detailed.

Comparing Table 5 with Table 3 signals again a crucial difference between the corrections by response styles versus overconfidence. The right panel demonstrates that explained variation after adding overconfidence as a predictor does not increase a lot. In contrast to the left panel: adding the two response styles variables has a substantial impact on explained variation.

## The epistemic origins of achievement emotions

Four weeks before measuring the achievement motivations embedded within the context of the mathematics and statistics learning tasks discussed in the middle of the course, learning emotions were measured within a more general context: learning for the course in general. Both this difference in timing and context suggest that epistemic emotions act as an antecedent for achievement emotions. To investigate if this antecedent-consequence relationship is invariant under correction for bias, we follow the same steps as in the previous subsection: first, decompose the predictor variables into bias component(s) and a bias-corrected component, and next investigate relationships with and without bias correction. Table 6 provides the decomposition of epistemic emotions measured by the EES instrument.

Response styles explain less variation in epistemic emotions than they do for achievement emotions; the effect of overconfidence is less clear. But similar to the case of achievement motivations, the effect of overconfidence explaining epistemic emotions is much smaller than the effect of the response styles.

The four regression equations relating the achievement emotions to the epistemic emotions are contained in Table 7.

Epistemic emotions explain 30% to 50% of the variation in achievement emotions. The pattern visible in the previous section, indicating that anxiety and hopelessness find a better explanation by academic control as well as response style or overconfidence, repeats with the current very different set of predictor variables. That is remarkable, since hopelessness is the single achievement emotion without a corresponding epistemic emotion, that in all of the other three regression equation absorbs most of the predictive power. In hopelessness, it is epistemic anxiety taking that role, with secondary roles for curiosity, surprise, confusion, frustration and boredom.

**Table 3. CVTAE model, standardised regressions coefficients of the model based on observed values.**

|       | ASC        | $R^2$ |
|-------|------------|-------|
| LAX   | -.538[***] | .290  |
| LBO   | -.307[***] | .094  |
| LHL   | -.709[***] | .502  |
| LJO   | .314[***]  | .099  |

[***]$p < .001$; [**]$p < .01$; [*]$p < .05$.

**Table 4. CVTAE model, standardised regressions coefficients of the model based on bias-corrected values.**

|  | ASCcorRS | $R^2$ | ASCcorConf | $R^2$ |
|---|---|---|---|---|
| LAXcor | -.355*** | .126 | -.511*** | .262 |
| LBOcor | -.094** | .009 | -.289*** | .084 |
| LHLcor | -.577*** | .333 | -.696*** | .484 |
| LJOcor | .289*** | .084 | .311*** | .097 |

***$p < .001$;

**$p < .01$; *$p < .05$.

The effect of correcting all emotion variables is displayed in two different tables: Table 8 when the correction is based on response styles, Table 9 for the overconfidence case.

Correcting for response styles has a substantial impact on the relationships between epistemic and achievement emotions: all explained variation values diminish in size, primarily because the role of the main predictor variable is diminished. The story of the overconfidence corrected regression models is different: due to limited collinearity of overconfidence with both types of emotions measurements, the prediction equations of boredom and enjoyment do not change by correcting measurements, whereas the prediction equations of anxiety and hopelessness do change slightly.

In the last modelling step, we add the bias term (either ERSpos and ERSneg or ΔConfidence) to the set of predictor variables and run the regressions with the observed versions of the achievement emotions. Tables 10 and 11 provide these regression outcomes.

Although the overconfidence variable is a significant predictor of all four achievement emotions, see Table 10, the decomposition of epistemic emotions into an overconfidence part and an orthogonal part does not increase predictive power. Explained variation is of the same order of magnitude as in Table 7. The story is again different for the response styles corrected measurements. Explained variation in Table 10 is substantially higher than that in Table 7. In all four regressions, the predictor with the largest beta is one of the response style variables. The general pattern we can distil from these regressions is that more extreme responses tend to increase the level of positive emotions, decrease the level of negative emotions. Both types of extreme responses have effects of similar directions in case of boredom and enjoyment, whereas, in the case of anxiety and hopelessness, the effect of the negative type of extreme response dominates the effect of the positive type.

## From self-report to course performance

Does bias in self-reports also influence objectively measured constructs, such as course performance? We investigate again using the AEQ questionnaire data but would have achieved similar outcomes by using other questionnaire data as predictors. As with the other analyses, we

**Table 5. CVTAE model, learning emotions regressed on bias-corrected ASC and bias terms.**

|  | ASCcorRS | ERSpos | ERSneg | $R^2$ | ASCcorConf | ΔConf | $R^2$ |
|---|---|---|---|---|---|---|---|
| LAX | -.268*** | .238*** | -.640*** | .502 | -.486*** | -.313*** | .334 |
| LBO | -.082*** | -.228*** | -.407*** | .247 | -.287*** | -.116*** | .096 |
| LHL | -.442*** | .058*** | -.648*** | .610 | -.670*** | -.270*** | .522 |
| LJO | .265*** | .376*** | .096*** | .229 | .310*** | .055 | .099 |

***$p < .001$; **$p < .01$; *$p < .05$.

**Table 6. Decomposition of the EES variables.**

|  | ERSpos | ERSneg | R² | RS | RScor | ΔConf | R² | Conf | Confcor |
|---|---|---|---|---|---|---|---|---|---|
| Curious | .384*** | .056* | .155 | .394 | .919 | .069** | .005 | .069 | .998 |
| Surprised | .232*** | -.251*** | .105 | .323 | .946 | -.013 | .000 | .013 | 1.000 |
| Confused | .129*** | -.511*** | .264 | .514 | .858 | -.271*** | .073 | .271 | .963 |
| Anxious | .249*** | -.558*** | .344 | .587 | .810 | -.336*** | .113 | .336 | .942 |
| Frustrated | .063* | -.493*** | .241 | .490 | .871 | -.271*** | .073 | .271 | .963 |
| Excited | .241*** | .093** | .071 | .267 | .964 | .080** | .006 | .080 | .997 |
| Bored | -.175*** | -.311*** | .138 | .372 | .928 | -.083** | .007 | .083 | .997 |

columns two, three and four provide the regression outcomes of the variables in column one on the two response styles: standardised regression coefficients and explained variation. Columns five and six provide the beta weights of the response style-based decomposition. In the second panel, the regression on ΔConfidence, and the decomposition based on overconfidence is provided;

***$p < .001$;

**$p < .01$;

*$p < .05$.

start with decomposing the course performance variables into a bias component and a component orthogonal to that bias. One would expect the bias component to be zero because these are not self-report data, but although the bias component tends to be smaller than in the self-report cases, it is nowhere zero, as is clear from Table 12.

The right panel of Table 12 tells that all relationships between the course performance variables and the overconfidence variable are insignificant from a practical point of view, in that explained variation is always less than 1%. Especially for the first two measures of course performance, Grade and MathExam, this is remarkable since overconfidence is defined as the difference between subjective and objective confidence, and both of these confidence constructs are defined by regression of MathExam on two predictor sets of self-reports respectively objective measures. Due to this inability of the overconfidence construct to explain variation in course outcome variables, we will leave it out of consideration in the remainder of this section, and the next section.

The left panel of Table 12 tells that the story of the response styles is very different. Explained variation is still not impressive for the quiz scores as intermediate course performance variables, but up to 10% for the final and total scores. In all cases, it is the negative extreme response style that dominates the prediction of course performance scores: students high on negative response styles score on average higher in exam and quizzes.

Regression equations explaining observed course performance variables form observed CVTAE variables indicates that explained variation is modest: 16% for the final course grade (see Table 13).

**Table 7. Epistemic emotions explaining achievement emotions, observed values: Standardised regression coefficients.**

|  | Curious | Surprise | Confusion | Anxiety | Frustration | Enjoyment | Boredom | R² |
|---|---|---|---|---|---|---|---|---|
| LAX | -.031 | .067** | .145*** | .490*** | .069 | -.077* | -.003 | .477 |
| LBO | -.111*** | .039 | .069 | -.042 | .031 | -.076* | .429*** | .319 |
| LHL | -.091** | .101*** | .109** | .333*** | .152*** | -.072* | .097*** | .422 |
| LJO | .237*** | .087** | -.111** | .039 | -.037 | .253*** | -.136*** | .387 |

***$p < .001$;

**$p < .01$;

*$p < .05$.

**Table 8. Epistemic emotions explaining achievement emotions, response styles corrected values.**

|          | CurCorRs | SurCorRs | ConCorRs | AnxCorRs | FruCorRs | ExcCorRs | BorCorRs | $R^2$ |
|----------|----------|----------|----------|----------|----------|----------|----------|-------|
| LAXCorRs | -.134*** | -.007    | .087*    | .341***  | -.021    | -.190*** | -.115*** | .273  |
| LBOCorRs | -.087*   | -.020    | .018     | -.128*** | -.020    | -.132*** | .349***  | .225  |
| LHLCorRs | -.140*** | .032     | .046     | .192***  | .080*    | -.176*** | -.014    | .208  |
| LJOCorRs | .140***  | .073**   | -.120*** | -.034    | -.068    | .234***  | -.136*** | .327  |

all response and predictor variables corrected for response styles;

***$p < .001$;

**$p < .01$;

*$p < .05$.

Main predictors are academic control and hopelessness. We see marked differences between the two topics of the course, mathematics and statistics. Hopelessness is a strong predictor for mathematics-related performance, but much less for statistics related performance. Causing a gap between explained variation in performance of both topics: $R^2$ measures are highest for the two math-related course performances.

Redoing the analysis with response styles corrected measures brings Table 14.

The explanatory power of all five of these regressions has decreased considerably: explained variation is less than half of the explained variation of the equations based on observed measures. The last step in the analysis refers to the explanation of observed performance variables by response styles corrected learning emotions plus the two bias terms themselves: see Table 15.

The explained variation visible from Table 15 is back to the level of the equations expressed in observed measures. The role of the main predictor has however shifted from the academic control variable to the negative extreme response scale: part of explained variation accounted for by academic control in Table 13, has shifted toward the negative extreme response style in Table 15.

## Self-report biases and trace and course performance variables

In this last step of the empirical analysis, we extend to the trace data and use these trace data to develop regression equations explaining the same five course performance variables as in the previous section. That is: these models are fully based on objective measures, both with regard to response variables, and predictor variables.

As a preliminary analysis, we decompose the trace variables exactly the way we did in the previous section, using both response styles and overconfidence as instrumental variables. The outcome is in Table 16.

**Table 9. Epistemic emotions explaining achievement emotions, overconfidence corrected values.**

|          | CurCorCf | SurCorCf | CfCorCf  | AnxCorCf | FruCorCf | ExcCorCf | BorCorCf | $R^2$ |
|----------|----------|----------|----------|----------|----------|----------|----------|-------|
| LAXCorCf | -.037    | .075**   | .160***  | .455***  | .054     | -.075*   | .007     | .432  |
| LBOCorCf | -.114*** | .042     | .079*    | -.057    | .017     | -.069    | .439***  | .318  |
| LHLCorCf | -.096    | .106     | .117     | .305     | .138     | -.074    | .108     | .385  |
| LJOCorCf | .241***  | .090***  | -.122*** | .035     | -.028    | .247***  | -.136*** | .388  |

all response and predictor variables corrected for overconfidence;

***$p < .001$;

**$p < .01$;

*$p < .05$.

**Table 10. Epistemic emotions explaining achievement emotions, response styles corrected values.**

|  | Cur CorRs | Sur CorRs | Con CorRs | Anx CorRs | Fru CorRs | Exc CorRs | Bor CorRs | ERS pos | ERS neg | R² |
|---|---|---|---|---|---|---|---|---|---|---|
| LAX | -.101*** | -.005*** | .066*** | .258*** | -.016*** | -.144*** | -.086*** | .239*** | -.633*** | .586 |
| LBO | -.076*** | -.018*** | .016*** | -.112*** | -.017*** | -.115*** | .305*** | -.227*** | -.401*** | .411 |
| LHL | -.107*** | .025*** | .035*** | .147*** | .061*** | -.135*** | -.010*** | .059*** | -.641*** | .537 |
| LJO | .128*** | .067*** | -.110*** | -.031*** | -.063*** | .215*** | -.125*** | .375** | .083*** | .434 |

all response and predictor variables corrected for response styles;

***$p < .001$;

**$p < .01$; *$p < .05$.

We included the corrections for overconfidence (right panel) to demonstrate that as in the course performance variables, overconfidence has no impact on the trace data collected from the learning processes. The largest $R^2$ equals 0.4% so that we will disregard this type of correction in the remainder of this section. Explained variation by the response styles is modest too, with the largest $R^2$ of 4.0%. From the three datasets incorporated in this study, it is clear that the learning activity trace data present the weakest relationship with the two bias factors we have composed. Another feature of interest is that the relationships of trace variables with the two response styles seem to be opposite to the relationships of performance and response styles: we find negative, rather than positive beta's for the negative extreme response scale, and positive, rather than absent, betas for the positive extreme response scale.

In the explanation of course performance variables by trace variables, we make use of the separate topic scores at hand and the topic-specific trace measures. That is why in Table 17, the predictors of the two mathematics course performance measures differ from the predictors of the statistics course performance measures, except for BBClicks. To address the collinearity within the set of trace variables, we had to remove the Math Sowiso Solutions variable that is highly collinear with Math Sowiso Attempts.

The main predictors in all four equations are the two product types of trace variables that represent the mastery levels achieved by the students in the two e-tutorials. The other trace variables derived from the two e-tutorials, all of the process type, all have negative or zero betas, although they are highly positively correlated with performance in bivariate relations. In combination with the mastery variable, the negative betas of NoAttempts, NoHints and Time tell that students who need more attempts, hints or time to reach the same level of mastery, score lower on average on course performance. Next, we observe that quiz performance is much better explained than exam performance, due to the close connection of quizzes and the e-tutorials.

**Table 11. Epistemic emotions explaining achievement emotions, overconfidence corrected values.**

|  | Cur CorCf | Sur CorCf | Cf CorCf | Anx CorCf | Fru CorCf | Exc CorCf | Bor CorCf | ΔConf | R² |
|---|---|---|---|---|---|---|---|---|---|
| LAX | -.035 | .071** | .152*** | .433*** | .051 | -.072* | .006 | -.313*** | .488 |
| LBO | -.113*** | .041 | .079* | -.057 | .017 | -.068 | .436*** | -.121*** | .327 |
| LHL | -.092** | .102*** | .113** | .293*** | .133*** | -.071* | .104*** | -.270*** | .430 |
| LJO | .240*** | .090*** | -.121*** | .035 | -.028 | .247*** | -.136*** | .060** | .390 |

all response and predictor variables corrected for overconfidence;

***$p < .001$;

**$p < .01$;

*$p < .05$.

**Table 12. Decomposition of the course performance variables.**

|           | ERSpos | ERSneg | R² | RS | RScor | ΔConf | R² | Conf | Confcor |
|-----------|--------|--------|-----|-----|-------|-------|-----|------|---------|
| Grade     | -.042  | .311***| .096| .310| .951  | -.008 | .000| .008 | 1.000   |
| MathExam  | -.062* | .278***| .078| .279| .960  | -.038 | .001| .038 | .999    |
| StatsExam | -.026  | .273***| .074| .272| .962  | .023  | .001| .023 | 1.000   |
| MathQuiz  | .037   | .237***| .059| .244| .970  | -.084**| .007| .084 | .996    |
| StatsQuiz | -.025  | .173***| .030| .172| .985  | .006  | .000| .006 | 1.000   |

columns two, three and four provide the regression outcomes of the variables in column one on the two response styles: standardised regression coefficients and explained variation. Columns five and six provide the beta weights of the response style-based decomposition. In the second panel, the regression on ΔConfidence, and the decomposition based on overconfidence is provided;

***$p < .001$;

**$p < .01$;

*$p < .05$.

Redoing the analysis with response styles corrected measures brings Table 18.

Regression equations in Table 18 are (practically) identical to those in Table 17, due to the circumstance that the response styles corrections have little impact on the trace variables. The last step of the analysis is the regression of the observed performance variables on the response style bias-corrected trace variables and the response styles. These outcomes, in Table 19, differ considerably from the two previous tables.

Because course performance variables contain a substantial response styles component, in contrast to the learning trace variables, we see that the explanation of course performance does improve adding the response styles to the predictor set. In some cases, that improvement is considerable: for the most difficult to explain performance measure, MathExam, the increase in explained variation is 40%.

## Discussion

The often-cited drawback of self-report data such as surveys and psychometric instruments is it biasedness: self-perceptions are seldom an accurate account of true measures. The question is: is this drawback unique for self-reports? To investigate this question, we constructed two different bias measures: one based on the differences between subjective and objective measures of confidence for learning in university, a type of under- and overconfidence construct, and the other based on extreme response styles. The selected response styles, both positive and negative extreme responses, make up a substantial part of all of the self-reported questionnaire variables, in size ranging between 7% and 43% of the explained variation. That indeed

**Table 13. Achievement emotions and academic control explaining course performance, observed values.**

|           | ASC     | LAX   | LBO     | LHL     | LJO    | R²   |
|-----------|---------|-------|---------|---------|--------|------|
| Grade     | .199*** | -.024 | -.023   | -.217***| -.034  | .161 |
| MathExam  | .126**  | -.047*| .089    | -.276***| .012   | .147 |
| StatsExam | .245*** | -.017 | -.085   | -.092*  | -.078* | .118 |
| MathQuiz  | .111**  | .003  | -.040   | -.244***| .086*  | .158 |
| StatsQuiz | .106*   | -.014 | -.113** | -.139*  | -.109**| .073 |

***$p < .001$;

**$p < .01$;

*$p < .05$.

**Table 14. Achievement emotions and academic control explaining course performance, response styles corrected values.**

|  | ASC CorRs | LAX CorRs | LBO CorRs | LHL CorRs | LJO CorRs | $R^2$ |
|---|---|---|---|---|---|---|
| GradeCorRs | .167*** | .028 | .002 | -.142** | .019 | .071 |
| MathExamCorRs | .103** | .005 | .099** | -.192*** | .056 | .074 |
| StatsExamCorRs | .203*** | .029 | -.053 | -.039 | -.026 | .049 |
| MathQuizCorRs | .092* | .016 | -.025 | -.181*** | .097** | .093 |
| StatsQuizCorRs | .092* | -.005 | -.098** | -.107* | -.098** | .036 |

***$p < .001$;

**$p < .01$;

*$p < .05$.

represents a considerable bias. However, the objectively measured performance measures allow the same decomposition and result in response styles contributions to explained variation in the lower end of that same range. Learning systems-based trace variables, both of product and process types, are most resistant to response styles, with the highest contribution to the explained variation of 4%. The role of the other type of bias we sought to operationalize, the overconfidence, is more modest. It stands out of course performance variables and trace variables, and is contained in some of the self-report variables, but nowhere with a variance contribution exceeding 10%.

Negative and positive extreme responses occur in different items: negative extreme response in items with a negative valence, where the scale mean is below the neutral value, and positive extreme response in positively valenced items, with scale means above the neutral value. Typically, the two extreme responses do not go together: items with a high ERSneg tend to have about zero ERSpos, such as learning helplessness, LHL (4) in Fig 1, and items that have high ERSpos tend to have about zero ERSneg. In Fig 1, that is the large cluster of variables on the right: since most items are positively valenced, there is a large group of academic motivations, goal setting and learning approaches variables ending up in that cluster on the right. There are a few exceptions to this pattern. Several instruments contain an anxiety-related scale, and three of these (LAX, Anxiety, AN) combine negative ERSneg weights with positive ERSpos weights. That is: if we wish to correct anxiety scores for response styles, true anxiety scores are lower than measured ones for those students with high ERSneg scores, and true anxiety score are higher than measured ones for those students with high ERSpos scores. If we look at Tables 2 and 6, we see that the correction induced by ERSneg is a consistent and strong one: in all negatively valenced constructs, we find that students with high ERSneg levels exaggerate their negative emotions, so a downward correction is required. Likewise, these students

**Table 15. Achievement emotions and academic control explaining course performance, response styles corrected predictor values.**

|  | ASC CorRs | LAX CorRs | LBO CorRs | LHL CorRs | LJO CorRs | ERS pos | ERS neg | $R^2$ |
|---|---|---|---|---|---|---|---|---|
| Grade | .158*** | .027 | .002 | -.134** | .018 | -.043 | .326*** | .168 |
| MathExam | .098** | .005 | .095** | -.184*** | .054 | -.062* | .291*** | .152 |
| StatsExam | .195*** | .028 | -.051 | -.038 | -.025 | -.028 | .285*** | .124 |
| MathQuiz | .088* | .016 | -.024 | -.174*** | .093** | .035 | .261*** | .158 |
| StatsQuiz | .090* | -.005 | -.096** | -.105* | -.096** | -.024 | .196*** | .073 |

***$p < .001$;

**$p < .01$;

*$p < .05$.

**Table 16. Decomposition of the trace variables.**

|  | ERSpos | ERSneg | $R^2$ | RS | RScor | ΔConf | $R^2$ | Conf | Confcor |
|---|---|---|---|---|---|---|---|---|---|
| BBClicks | .139*** | .032 | .021 | .146 | .989 | -.035 | .001 | .035 | .999 |
| MathMastery | .107*** | .027 | .013 | .127 | .992 | -.066* | .004 | .072 | .998 |
| MathAttempts | .139*** | -.111*** | .029 | .167 | .986 | -.024 | .001 | .026 | 1.000 |
| MathSolutions | .092** | -.187*** | .040 | .190 | .980 | .002 | .000 | .000 | 1.000 |
| MathHints | .049 | -.013 | .002 | .050 | .999 | .029 | .001 | .028 | 1.000 |
| MathTime | .118 | -.054 | .016 | .127 | .992 | -.036 | .001 | .037 | .999 |
| StatsMastery | .116 | .046 | .017 | .129 | .992 | -.054 | .003 | .054 | .999 |
| StatsAttempts | .122*** | -.020 | .015 | .121 | .993 | -.054 | .003 | .054 | .999 |
| StatsTime | .105*** | -.065* | .014 | .117 | .993 | -.047 | .002 | .047 | .999 |

columns two, three and four provide the regression outcomes of the variables in column one on the two response styles: standardised regression coefficients and explained variation. Columns five and six provide the beta weights of the response style-based decomposition. In the second panel, the regression on ΔConfidence, and the decomposition based on overconfidence is provided;

***$p < .001$;

**$p < .01$;

*$p < .05$.

undervalue their positive emotions, so an upward correction is demanded. The role of ERSpos is less unambiguous and not uniquely determined by the valence of the scale. In several scales, we find that an upward correction is required for students high in ERSpos scores: their anxiety levels are higher than measured, but their enjoyment and curiosity levels too. The exception is in boredom, both epistemic and achievement type: students who tend to provide extreme positive responses, exaggerate their boredom levels, calling for a downward correction.

The patterns induced by overconfidence mirror those of the negative extreme response style, but at a smaller scale. Overconfidence increases the level of the constructs with a positive valence, as academic control and enjoyment, and decreases the levels of negative emotions: anxiety of several types, hopelessness, frustration and confusion. Correcting for overconfidence will thus imply a downward correction of these positively valenced constructs and an upward correction for the negatively valenced constructs.

Course performance variables follow most of the patterns of the positively valenced self-report scales, in that we find consistent, strong ERSneg contributions. That is: expected performance levels of students with high ERSneg scores should be corrected in an upward direction. Remarkably, no correction for ERSpos scores is needed, what results in the course performance variables clustering together along the positive part of the vertical axis in Fig 2.

**Table 17. Trace variables from learning systems explaining course performance, observed values.**

|  | BB Clicks | Math Sowiso Mastery | Math Sowiso Attempts | Math Sowiso Hints | Math Sowiso Time | $R^2$ |
|---|---|---|---|---|---|---|
| MathExam | .080** | .637*** | -.551*** | -.067* | -.006 | .172 |
| MathQuiz | .129*** | .721*** | -.281*** | -.038 | .032 | .336 |
|  | BB Clicks | Stats MSL Mastery | Stats MSL Attempts |  | Stats MSL Time |  |
| StatsExam | .061* | .864*** | -.522*** |  | -.127*** | .212 |
| StatsQuiz | .039 | 1.040*** | -.428*** |  | -.140*** | .426 |

***$p < .001$;

**$p < .01$;

*$p < .05$.

**Table 18. Trace variables from learning systems explaining course performance, response styles corrected values.**

|  | BB ClicksCorRs | Math Sowiso MastCorRs | Math Sowiso AttCorRs | Math Sowiso HintsCorRs | Math Sowiso TimeCorRs | $R^2$ |
|---|---|---|---|---|---|---|
| MathExamCorRs | .084** | .653*** | -.546*** | -.072* | .016 | .169 |
| MathQuizCorRs | .100*** | .730*** | -.264*** | -.029 | .042 | .343 |
|  | BB ClicksCorRs | Stats MSL MastCorRs | Stats MSL AttCorRs |  | Stats MSL TimeCorRs |  |
| StatsExamCorRs | .061* | .864*** | -.522*** |  | -.127*** | .212 |
| StatsQuizCorRS | .039 | 1.040*** | -.428*** |  | -.140*** | .426 |

***$p < .001$;

**$p < .01$;

*$p < .05$.

Together with Academic control, ASC (5), Cognitive competence (34) and Affect (35), the three variables expressing perceived self-efficacy.

In Fig 2, the cluster nearest to the origin is that of the learning activity trace variables. They are least biased, and thus need no more than a small correction, given that students with high positive extremes tend to have slightly higher average activity levels.

In the literature on 'fundamental validity problems' [2, 3], the individual reference problem and the memory problem can both explain the existence of response styles like differences in answer patterns between students. If the absence of such answer patterns is taken as a definition of the true level of measurement, then it is clear that all of the self-reports, as well as course performance variables, represent biased constructs, and that the decomposition of these variables into a response style component and a component orthogonal to that, is one of taking the bias out. However, to make validity into a meaningful concept, it has to be criterion-related. In educational research, that criterion is that it helps understanding educational theories: theories that relate multiple educational concepts measured with different instruments, or theories that relate such concepts with the outcomes of educational processes. If that is the main criterion, then our definition of validity and bias should change. A valid instrument is then an instrument that contains such typical person-specific response patterns, and bias is now defined as the incapability of the instrument to account for such patterns. In the context of our application: it is the self-report and course performance data that represent the unbiased parts of our data collection, since we aim to investigate the empirical model of the control-value theory of achievement emotions (CVTAE) and its contribution in the explanation of course outcomes, whereas our trace variables represent the biased part of our data collection, due to its inability to account for these typical personal patterns in the data that determine our criterion.

**Table 19. Trace variables from learning systems explaining course performance, confidence difference corrected predictor values.**

|  | BBClicks CorRs | MathMast CorRs | MathAtt CorRs | MathHints CorRs | MathTimeCorRs | ERS pos | ERS neg | $R^2$ |
|---|---|---|---|---|---|---|---|---|
| MathExam | .080** | .623*** | -.521*** | -.069* | .015* | -.067* | .292*** | .242 |
| MathQuiz | .097*** | .707*** | -.255*** | -.028 | .040 | .028 | .243*** | .384 |
|  | BBClicks CorRs | StatsMast CorRs | StatsAtt CorRs |  | StatsTime CorRs | ERS pos2 | ERS neg2 | $R^2$ |
| StatsExam | .055* | .800*** | -.100*** |  | -.462 | -.023 | .283*** | .273 |
| StatsQuiz | .031 | 1.007*** | -.126*** |  | -.402*** | -.030 | .195*** | .453 |

***$p < .001$;

**$p < .01$;

*$p < .05$.

The alternative modelling approach, with response style correction of the variables as a first step of the analysis, seems not being very practical. That approach would require the correction of all measured variables in the analysis the way we did it in this article. Take for an example, the very simple CVTAE relationship explaining helplessness from academic control. Rather than having one relationship, we would arrive at three:

$$LHL = -0.709^{***} * ASC; R^2 = 0.502$$

$$LHLcor = -0.577^{***} * ASCcor; R^2 = 0.333$$

$$LHL = -0.442^{***} * ASCcor + 0.058^{***} * ERSpos - 0.648^{***} * ERSneg; R^2 = 0.610$$

Is it the second equation that we prefer? It has eliminated the impact of response styles, at least those we distinguished, but says nothing about other potential biases. The third equation has the advantage that it allows an impression of the impact of response styles, but it is in unattractive, non-parsimonious format. One needs all three expressions, because the first two help understand the extent to which helplessness and academic control share the same response styles, and the third one provides the decomposition into response style or not. However, the problem is that the response style is only one source of bias. In this example, confidence difference brings the second type of bias, accounting for 7% of the variation. Adding this second correction or any further correction one can think of, would add explanatory power, but make for an explanation most obviously lacking any parsimony, without the guarantee that all bias sources are covered. It is therefore that we prefer the first formulation of the three equations, knowing that this choice sacrifices at least 10% of the explained variation, resulting from the circumstance that helplessness carries a larger response style component than academic control can account for.

The outcome of this study that connects with all our previous research [14–18] is that we once more discovered how "dangerous" learning activity trace data of process type can be. In this study, we included NoAttempts, NoSolutions, NoHints, and TimeOnTask as examples of such process variables. All these variables demonstrate strong positive bivariate relationships with all of the learning performance variables, telling the simple message: the more active the student, the higher the expected learning outcomes. Nevertheless, that simple message is deceptive: as soon as we add a covariate of product type, such as Mastery in the learning tool, the role of the process predictors changes radically: relationships become negative or vanish. In itself not surprising, and easily explained by a second simple mechanism. The student who needs to consult more worked-out examples (Solutions), the student who needs more Hints, the student who needs more Attempts, the student who needs more TimeOnTask, than another student to reach the same level of Mastery, is learning less efficiently, and therefore predicted to achieve lower course performance scores on average. The obvious way out of this problem is finding the causes of these efficiency differences and correct for these factors. That is no easy way to go; although having access to a huge database of personal characteristics of students, none of these qualified as a proper predictor of learning efficiency. Prior education, diagnostic entry test scores and other variables of this type all explain a small part of these efficiency differences, but no more than that.

Reflecting on our research questions: we do find that self-report survey data and course performance data largely reflect the conceptualisations of the constructs we intended to find in our models. Both types of constructs contain response style type of components of modest to a substantial size. These might be regarded as components of bias, and contrast to the trace variables that lack these bias components. However, is it reasonable to make these traces the

standards of our educational theories? When designing models, we hardly ever will do so with the prime aim of explaining levels of traces of learning activity. The majority of our models seek to understand the outcomes of learning processes or investigate the relationships between social-cognitive antecedents of these learning outcomes. Therefore, if these modelling aims define our standards, the bias is at the side of the trace variables in that they need to be corrected to include the stable response style patterns that characterise all other variables in our models.

These differences in response style patterns do not necessarily constrain analytical choices. If a sufficiently rich set of self-report data is available, as in our application, we can make a reliable decomposition of all variables in the analysis. Models that build on such a decomposition have the advantage of high predictive power, against the disadvantage of being less parsimonious, more difficult to interpret. If we prefer to stick with parsimonious models that apply measured variables only without correction, we are indeed restrained in our analytical choices. In our case, that restriction comes down to a limitation of the role that trace variables can play in the explanation of other types of variables, due to their incapability to catch the response style components.

The inclusion of response styles as separate explanatory factors does change the interpretation of models somewhat. In a manner that is quite intuitive: if we isolate the response styles components from the achievement emotions, as in Table 15, the achievement emotions will lose part of their predictive power in favour of the response styles. That is exactly what happens in the comparison between Tables 13 and 15: it is still academic control, ASC, and helplessness, LHL, that predict the several course performance categories, with a positive beta for ASC and a negative beta for LHL, but the absolute size of these betas are diminished. That predictive power is now absorbed by ERSneg. This finding can be generalised: when we estimate models of learning processes that are formulated in terms of variables that share a common component, such as a response style or any other 'bias', we will find inflated estimates caused by the circumstance that the same bias component is part of both response and predictors. Any predictor that is free of that bias component will also be free of such an inflated estimate.

## Limitations and future directions

In our context, we find that it is the trace type of data that stands out in the sense that these data cannot be easily integrated with self-report and course performance data. That is a robust outcome: the same data-rich context used in this study has been investigated in more than ten years of learning analytics research, always with that same conclusion. Strong heterogeneity in our population may be part of the explanation of why the trace variables are so out of synch with the other measured constructs. High levels of learning activity may signal a student who likes doing the subject and is very good at it or a very conscientious student but may also be an indicator for extra learning efforts required to compensate low proficiency levels at the start of the course. Where the heterogeneous population benefits, in general, most model building endeavours, it clearly limits the analysis of the learning activity to learning outcomes relationship. If this analysis could be repeated in a more homogeneous sample, we might have found more stable roles for the online trace variables. However, it would not help to solve the other issue: by not being able to capture response patterns characterising questionnaire and learning outcome data, their role in empirical models based on multi-modal data is problematic.

Heterogeneity in our sample is not the only difference with other studies. Quite a lot of studies are based on experimental design, with limited numbers of participating students and focussing on learning activities of limited intensity. For instance, the Zhou and Winne study [4] is based on 95 students in a one-hour experimental session, and the Fincham study [43] on

230 students participating in one of three different MOOCs. These MOOCs lasted five to ten weeks, but per active week, students watched an average of less than one video and submitted between one and two problems. In contrast, our study (and previous ones) focus on learning activities with far higher intensity. During our eight-week course, students do, on average, 760 problem-solving attempts in the math e-tutorial and 210 attempts in the stats e-tutorial, in total more than 120 attempts per week. Given that the number of problems offered per week fluctuates between 40 and 80, a substantial part of these attempts represents repeated attempts, where the need to repeat attempts differs strongly from student to student. Therefore, it is not unlikely that the difference in the role of trace variables of process type play is a consequence of investigation of learning in a small-scale experimental design, versus participating in intensive activities in an authentic learning context. More research on the role of the learning context is needed to answer such questions.

A third option to extend this study is turning it into a multiverse analysis [38] by investigating alternative data sets and alternative statistical methods to validate our findings in different contexts. The application of robust regression methods is a prime candidate of such alternative statistical method, as well as the application of slider rating scales in the administration of the several self-report instruments.

The last topic of future research refers to the mechanism at work that might explain the relationships between response styles and learning outcome variables, or trace variables of product type. Potential antecedents of response styles, such as cultural factors or gender, have been researched [8, 29]. But these studies are not of much help in explaining why response styles based on questionnaire data do show up in other types of data, like performance data and trace data of product type. More research is needed here.

## Conclusions

The large-scale introduction of technology-enhanced learning environments has had a huge impact on education as well as educational research. Questionnaire data, long time being the main source of empirical studies of learning and teaching, lost its prominent position to online data collected as digital traces of learning processes. Because this online data, using the term that is used in the area of metacognitive research, refers to data that is collected during the learning process itself, by following the student in all learning activity steps. Trace data of process type collected by technology-enhanced learning systems is an excellent example of such on-line data (whereas trace data of product type is, in fact, part of the off-line data, because it refers to reaching a state of mastery, what is not a dynamic process). In that debate, the outcome is invariable that subjective off-line data is inferior to objective on-line data. In our learning analytics research, where the learning outcome is typically the response variable that is explained and predicted, we find the opposite conclusion invariably: it is the on-line trace data of process type that is inferior to the off-line data and the online trace data of product type. In the generation of explanatory models, the role of process type of trace variables is quite unstable, depending strongly on the covariates in the model. Regression betas can become insignificant or switch signs after adding covariates, especially if these are of product type.

In contrast, product type of trace variables tends to play stable roles, with little disturbance of the addition of covariates. The critique towards data of self-report type, too stable and too trait-oriented [37] is reversed in this application: it is the trace data of process type, even after aggregation over the full course period, which lacks the stability to act as a reliable predictor. That is what one decade of learning analytics research brought the authors as insight: be very careful with online data of process type, put more trust in online data of product type, complemented with survey data.

## Supporting information

**S1 Appendix. Instruments of self-report surveys [44–49].**
(PDF)

**S2 Appendix. Descriptive statistics of all variables in the study.**
(PDF)

**S3 Appendix. Correlations of mean response styles measures.**
(PDF)

## Author Contributions

**Conceptualization:** Dirk Tempelaar, Bart Rienties.

**Data curation:** Dirk Tempelaar, Quan Nguyen.

**Formal analysis:** Dirk Tempelaar, Quan Nguyen.

**Investigation:** Dirk Tempelaar, Bart Rienties.

**Methodology:** Dirk Tempelaar, Bart Rienties, Quan Nguyen.

**Writing – original draft:** Dirk Tempelaar, Bart Rienties, Quan Nguyen.

**Writing – review & editing:** Dirk Tempelaar, Bart Rienties, Quan Nguyen.

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
