## [Decision Letter · Decision Letter 0]

25 Mar 2020

PONE-D-20-04502

Subjective data, objective data and the role of bias in predictive modelling: lessons from a dispositional learning analytics application

PLOS ONE

Dear Dr. Tempelaar,

Thank you for submitting your manuscript to PLOS ONE. After careful consideration, we feel that it has merit but does not fully meet PLOS ONE’s publication criteria as it currently stands. Therefore, we invite you to submit a revised version of the manuscript that addresses the points raised during the review process.

Both reviewers appreciated your manuscript and recognized its importance and contribution in understanding how the bias of the different types of data affects the quality of predictive models. Indeed, the effect of bias on learning analytics models is a very timely issue and instrumental in the way we use LA in the future. 

However, R1, who is highly knowledgeable in statistical modelling, indicated several issues in the way your analysis was presented and reported. I think you can address most of those issues and would be looking to receive the revised manuscript. 

We would appreciate receiving your revised manuscript by May 09 2020 11:59PM. To enhance the reproducibility of your results, we recommend that if applicable you deposit your laboratory protocols in protocols.io, where a protocol can be assigned its own identifier (DOI) such that it can be cited independently in the future. For instructions see: http://journals.plos.org/plosone/s/submission-guidelines#loc-laboratory-protocols

We look forward to receiving your revised manuscript.

Kind regards,

Vitomir Kovanovic, Ph.D.

Academic Editor

PLOS ONE

Journal Requirements:

Reviewers' comments:

Reviewer's Responses to Questions

**Comments to the Author**

1. Is the manuscript technically sound, and do the data support the conclusions?

Reviewer #1: Yes

Reviewer #2: Yes

2. Has the statistical analysis been performed appropriately and rigorously? 

Reviewer #1: Yes

Reviewer #2: Yes

3. Have the authors made all data underlying the findings in their manuscript fully available?

Reviewer #1: No

Reviewer #2: Yes

4. Is the manuscript presented in an intelligible fashion and written in standard English?

Reviewer #1: Yes

Reviewer #2: Yes

5. Review Comments to the Author

Reviewer #1: Here are my comments on the manuscript

1. The authors mention seven-point Likert scales. Was each of the points in the scale paired with a label or only the anchors of the scale were labelled? This is essential as the former case requires multinomial (polytomous) logistic regression whereas the latter can be analysed via (linear) regression.

2. Data analysis section: the author mention that different statistical options were done but there is no justification to do so. I recommend considering labelling it ‘multiverse analyses’. Arguments for this approach are given in Steegen et al (2016).

3. N-points Likert scales have been in use since 1932 when Rensis Likert proposed them. But there’s been progress as to their use. One approach is that the more points in the scale, the larger the control of Type I errors (see Sangthong, 2020). A better approach would be to make use of slider rating scales in that it produces bounded and continuous distributions (e.g. values between 1 and 7 but that can take, say, two decimal places). These distributions enable more fined grained results. These issues need to me considered in the discussion.

4. The authors speak of ‘endogeneity’ (largely used in econometrics and understood as an independent variable being correlated with the error term); is this concept seems to be made analogous to ‘meditation analysis’? definition of both and their distinction needs to be made explicit via a footnote.

5. Results section, table 2: the authors use alpha coefficients. I suggest reading McNeish (2018) and reconsider using another coefficient.

6. Table 2: distributions appear to be non-normal (based on the skewness estimations); in non-normal cases, the mean isn’t the most unbiased estimator of location. A simpler and robust alternative is the median or 20% trimmed mean.

7. The authors performed several analyses and reported them via several tables. I wonder if there is a way those tables can be sent to a supplementary material so the main message is given in the text. Perhaps an alternative is to summarise those tables in a few graphs.

8. Conclusions section: the last sentence was incomplete.

Extra comments: the authors used ordinary linear regression (or OLS). Most likely data don’t meet assumptions needed for those models to perform well. If there is true interest in estimating unbiased beta weights, I recommend using robust linear regression methods (see for example Wilcox, 2017).

Finally statistical codes (e.g. R, SAS, etc) and data should be made available via a URL that takes the reader to a repository (e.g. figshare).

References

Steegen, S. et al (2016). Increasing Transparency Through a Multiverse Analysis. Perspectives on Psychological Science, 11(5), 702-712

Sangthong, M. (2020). The Effect of the Likert Point Scale and Sample Size on the Efficiency of Parametric and Nonparametric Tests. Thailand Statistician, 18 (1)

McNeish, D. (2018). Thanks coefficient alpha, we'll take it from here. Psychological Methods, 23(3):412-433

Wilcox, R. R. (2017). Introduction to Robust Estimation and Hypothesis Testing, 4th Ed. Academic Press.

Reviewer #2: This is a very interesting and well-written paper. It examines the biases inherent in different forms of data often used for predictive modelling in learning analytics in a very thorough and thought-provoking way.

The authors set out a solid case for this approach in the beginning of the paper, highlighting the importance of these discussions within the wider learning analytics discourse. One slight issue on the second page of the main article - it is stated that "In the educational psychology literature, it is widely acknowledged that although questionnaires…", but the two references given at the end of the sentence are based in the marketing domain. Having said that, the incorporation of response styles from the marketing discipline into this LA conversation provides an valuable perspective on this ongoing issue in LA research.

Whilst the methods and analysis involved in this study are substantial and complicated, the authors have presented these in a clear fashion with helpful descriptions and justification throughout.

In Table 1 - it appeared that the variable name and variable acronyms have been swapped between columns in the rows from MEMO down - is that right?

The implications of this substantial amount of work are well distilled into a useful discussion and thoughts on future directions at the end of the paper. It was refreshing to see the straight-forward presentation of what this means for how we can approach the use of different forms of data in future LA work.

The paper contains a few grammatical errors and missing words - so would benefit from a thorough proofread prior to publication.

6. PLOS authors have the option to publish the peer review history of their article (what does this mean?). If published, this will include your full peer review and any attached files.

Reviewer #1: Yes: Fernando Marmolejo-Ramos

Reviewer #2: No

---

## [Author Response · Author response to Decision Letter 0]

7 Apr 2020

(please see the uploaded document that contains this information in tabular format)

Authors’ response

Dear academic editor, dear Vitomir,

We appreciate the feedback and recommendations we received from you and the two reviewers, specifically those from Fernandez. We have addressed all of them in the revised version of the paper and elaborate them in this rebuttal.

The data, the MPlus and SPSS codes, and the main components of the output are archived in DANS, the Data Archiving and Networked Services of the NOW, the Dutch organization of scientific research. DANS is an open access resource. The draft version of this archive, labelled Tempelaar, D, 2020, "Replication Data for PlosOne 2020 manuscript Tempelaar ea", has received the unique handle: https://hdl.handle.net/10411/YAF7CJ, DataverseNL, DRAFT VERSION

Upon receiving the DOI of the publication, the archive can be finalized.

We are very grateful for the detailed feedback from Reviewer 1. Our research applies eight different self-report instruments, as reported in Table 1. These eight instruments were designed with different types of Likert response scales: different length and different labels. To simplify and ease the responses to these instruments, we applied the same length for all instruments, and indeed fixed the labels to include the three anchors only: the negative pole, the neutral anchor and the positive pole. This explanation is added.

We are new to this development of ‘multiverse analysis’ but can see the merits of this approach. We agree that our study would have been a perfect situation to apply this new approach. We have included a reference to Steegen’s work (she is nearly my neighbor, being associated to Leuven University) in the Data analysis section, and a further reference in the ‘future directions’ section since this suggestion is indeed worthwhile to adopt in future studies (it is too wide-ranging to adopt it in the current study). We hope that the main goal of multiverse analysis, the validation of empirical research outcomes by using other statistical methods or other versions of data sets, is opened by other researchers applying response styles. 

The Sangthong (2020) paper was new for us, but we were happy to see that his main conclusions, use long Likert scales, 7-point or 10-point, and use sample sizes of at least 100, are satisfied in our research. We have added this reference and a short discussion of the role of the length of the scale and the sample size to the data analysis section. We have added the reference to the slider rating scales to the section of future research, as part of the multiverse analysis.

Yes, that is correct. Since PlosOne forbids the use of footnotes, we have included the following explanation, within brackets, in the text: (the external validation of our extreme response measures, by investigating their role in the explanation of responses to an instrument not included in the calculation of extreme response measures).

We have read the McNeish article: another new area for us. We decided to address this comment by adding the Omega measure to Table 2, which now contains both alpha and omega measures. The reason to complement with omega, rather than substitute by omega, is that researchers in the domain of empirical educational studies are an important audience for us. These people, like us, are raised in the world of APA style formats, that require the description of alpha measures when using survey data. Replacing the alpha measures by omega measures, certainly when such measures are always ‘more positive’, will be a suspicious act in the eyes of such researchers, and we should try to avoid that impression. Therefore, we have added the Omega measures to Table 2, and have included in the main text references to the McNeish article and the Bandalos book from which we took the MPlus code to calculate the Omage measures.

By the way: except for the two ERSneg measures, all other variables have Omega measures that are essentially equal to the alpha measures.

We have added median scores to Table 1 (now Appendix B) and Table 2 (now Table 1). Probably due to the large sample size, differences between median scores and mean scores are quite small (except for some of the trace measures, contained in Appendix B).

Yes, we agree and already when giving shape to the first version, we have experimented with alternative displays of the information contained in the tables, including graphics. But with each attempt, we judged it to be less satisfying than the simple but dull tables. The problem is that all tables contain data with different ranges, so that we either arrive at graphs with large parts of white space, or we need to do nasty transforms of the data to get all of them on a comparable range. The other problem is that all Tables 3-20 contain similar data consisting of regression betas and R2 values, so there is no good reason the turn some into graphs leaving the others as tables.

In the end, we decided to place Table 1 in a statistical appendix: it is by far the largest table, spanning two pages, and it is the least important table for understanding the main outcomes of our study.

Sentence has been completed.

Thank you for this suggestion. A robust regression is an appropriate alternative to OLS in situations where outliers can heavily bias the beta estimation. However, given the large sample (N=1080), and relatively small skewness values, OLS can produce reasonable estimates in this case. 

We have noted robust regression as a potential direction for future research in the discussion. 

We are also extremely grateful to the feedback from Reviewer 2. You are right: we should not reference here to educational research, but to research in the survey method domain. Revised in that manner.

Right again. Part of the 1st and 2nd columns of Table 1, now Appendix B, got mixed up. We corrected this.

Yes, the revised paper was proofread by two readers.

---

## [Decision Letter · Decision Letter 1]

18 May 2020

Subjective data, objective data and the role of bias in predictive modelling: lessons from a dispositional learning analytics application

PONE-D-20-04502R1

Dear Dr. Tempelaar,

We are pleased to inform you that your manuscript has been judged scientifically suitable for publication and will be formally accepted for publication once it complies with all outstanding technical requirements.

With kind regards,

Vitomir Kovanovic, Ph.D.

Academic Editor

PLOS ONE

Additional Editor Comments (optional):

Reviewers' comments:

Reviewer's Responses to Questions

**Comments to the Author**

1. If the authors have adequately addressed your comments raised in a previous round of review and you feel that this manuscript is now acceptable for publication, you may indicate that here to bypass the “Comments to the Author” section, enter your conflict of interest statement in the “Confidential to Editor” section, and submit your "Accept" recommendation.

Reviewer #1: All comments have been addressed

Reviewer #2: All comments have been addressed

2. Is the manuscript technically sound, and do the data support the conclusions?

Reviewer #1: Yes

Reviewer #2: (No Response)

3. Has the statistical analysis been performed appropriately and rigorously? 

Reviewer #1: Yes

Reviewer #2: (No Response)

4. Have the authors made all data underlying the findings in their manuscript fully available?

Reviewer #1: Yes

Reviewer #2: (No Response)

5. Is the manuscript presented in an intelligible fashion and written in standard English?

Reviewer #1: Yes

Reviewer #2: (No Response)

6. Review Comments to the Author

Reviewer #1: I'm glad to read my comments relating to the methods/stats were attended to. Starting to become familiar with robust techniques will enable to obtain unbiased estimators readily used in explanatory and predictive research.

Reviewer #2: (No Response)

7. PLOS authors have the option to publish the peer review history of their article (what does this mean?). If published, this will include your full peer review and any attached files.

Reviewer #1: Yes: Fernando Marmolejo-Ramos

Reviewer #2: No

---

## [Editor Report · Acceptance letter]

27 May 2020

PONE-D-20-04502R1 

Subjective data, objective data and the role of bias in predictive modelling: lessons from a dispositional learning analytics application 

Dear Dr. Tempelaar:

I am pleased to inform you that your manuscript has been deemed suitable for publication in PLOS ONE. Congratulations! Your manuscript is now with our production department. 

With kind regards,

on behalf of

Dr. Vitomir Kovanovic 

Academic Editor

PLOS ONE